# Longitudinal Enrichment of Imaging Biomarker Representations for Improved Alzheimer's Disease Diagnosis

## Abstract

Longitudinal data is often available inconsistently across individuals resulting in ignoring of additionally available data. Alzheimer's Disease (AD) is a progressive disease that affects over 5 million patients in the US alone, and is the 6th leading cause of death. Early detection of AD can significantly improve or extend a patient's life so it is critical to use all available information about patients.

We propose an unsupervised method to learn a consistent representation by utilizing inconsistent data through minimizing the ratio of $p$-Order Principal Components Analysis (PCA) and Locality Preserving Projections (LPP). Our method's representation can outperform the use of consistent data alone and does not require the use of complex tensor-specific approaches. We run experiments on patient data from the Alzheimer's Disease Neuroimaging Initiative (ADNI), which consists of inconsistent data, to predict patients' diagnosis. Our results show that we can utilize additional data about patients, even when they are inconsistent, to improve their diagnosis accuracy thereby improving or lengthening their lifetime. We also identify key biomarkers predictive of AD consistent with clinical research.

## 1 Introduction

The use of longitudinal data, or data of the same nature collected over time, in machine learning is a heavily studied field. Medical data traditionally comes in this format and so machine learning applications on longitudinal data are inherently effective in patient disease diagnosis.

Alzheimers Disease (AD) affects millions of Americans each year and its seriousness is evident with it being the 6th leading cause of death in the US, see (Association et al., 2018). As time passes, AD patients' memory and mental abilities decline. Medical research in (Logsdon et al., 2007; Leifer, 2003) has shown that the early detection of AD can vastly improve patients' lives and potentially prevent the disease becoming deadly. Since AD's progression is important to track, subjects generally conduct multiple medical scans to monitor the disease's progression. Two general problems arise with longitudinal data; complexity of tensor methods and inconsistency of data over time.

If we consider each patient to have a vector of features at a single time point, the patient is represented as a matrix when viewed over time and a set of patients as a tensor. Representing the problem as a tensor can cause difficulty when predicting the diagnosis of a patient due to the missing data. This problem is heavily studied with (Wang et al., 2012; Brand et al., 2018; Lu et al., 2018) as example approaches which, despite their effectiveness, present an added complexity to the problem.

As the number of samples and time-steps increases, we find inconsistency across the data since samples will have some missing data at varying time points. This is exceedingly common in medical data due to the difficulty and cost of ensuring all patients conduct scans at the same time. Many approaches can be applied to handle missing data, however researchers most commonly select the subset of patients who have consistently available data. This approach discards a substantial amount of available data that could be used to provide value in predicting patients' diagnosis.

Our approach is to learn a new representation for each patient that incorporates all their available data, while having consistent dimensions with other patients. This consistent representation allows

us to apply classical supervised learning methods to our problem and can be learned from inconsistent longitudinal data. In our chosen application, we focus AD diagnosis from MRI scans where all subjects received an initial scan that we refer to as the baseline. Subjects returned to receive more scans at varying time-steps such as 6, 12, and 24 months later but not all subjects conducted a scan at all the time points. We enrich the baseline representation using additional scans of each patient by balancing between the global patterns and pairwise local patterns across the patient's data.

For each patient $i$, we have a vector of features representing their baseline scan $\mathbf{x}_i$ in $\mathbb{R}^d$ and a matrix of additional scans that they received $\mathbf{X}_i$ in $\mathbb{R}^{d \times n_i}$ where $n_i$ is the number of their additional scans. Our method learns a new representation for a patient, $\mathbf{y}_i$ that is enriched by the additional longitudinal data available for that patient, $\mathbf{X}_i$. We do so by learning a projection $\mathbf{W}$ in $\mathbb{R}^{d \times r}$ from the additionally available data, where $r$ is a hyperparameter. Figure 1 shows a visual representation of the enrichment we apply to the data. We calculate each individual patient separately so for notation simplicity we will discard the $i$ subscript when there is no interaction across patients. The enriched representation is calculated as,

$$\mathbf{y} = \mathbf{W}^T \mathbf{x}. \tag{1}$$

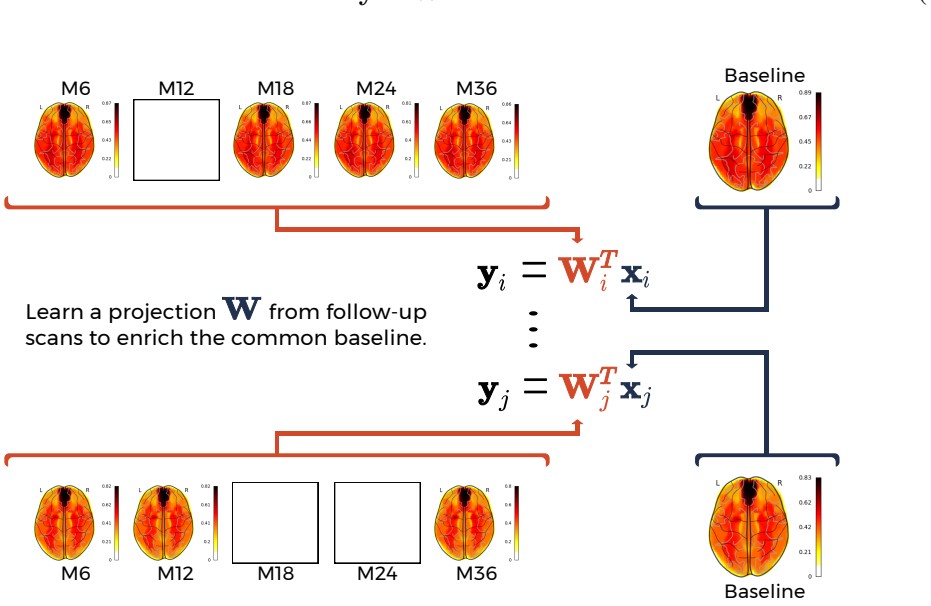

Figure 1: Enriching consistent patient data with additional data despite inconsistencies.

Another relevant aspect of data is how we consider outliers. Outliers can heavily impact the learned models, especially when we utilize $\ell_2$-norm distances such as (Wang et al., 2014). Many robust methods, (Wang et al., 2014), have been developed to manage the impact of outliers especially when using the squared $\ell_2$-norm distance. Some have chosen to use the $\ell_0$-norm, $\ell_1$-norm, or $\ell_{2,1}$ norm as a way to mitigate outliers' disproportionate effect. In our method, we replace the squared order of the distances with a parameter $p$ where $0 < p \leq 2$. Having $p$ as a parameter that includes 1&2 allows us to control and tune the impact of outliers in our model in a data-specific way. We refer to $p$ as the *outlier sensitivity*. Possessing fine control over the model's sensitivity to outliers can be a key performance advantage in various applications such as medical diagnosis.

## 2  OBJECTIVE

The general objective for our method is to learn a useful projection from additional longitudinal data and then apply the projection to enrich the baseline data. The projection is learned for each patient independently. The $i$ and $j$ subscripts reference the individual scans that make up the additional data for each target patient. The projection is comprised of both global and local patterns learned from the additional data. To capture the global patterns, Principal Component Analysis (PCA) creates a reduced representation of the longitudinal data that maximizes the explained variance across the

data. PCA is a maximization of,

$$\mathcal{J}_{PCA} = \sum_{i=1}^{n_i} ||\mathbf{W}^T \mathbf{x}_i||_2^2, \qquad s.t. \mathbf{W}^T \mathbf{W} = \mathbf{I}. \tag{2}$$

In addition to accounting for global patterns within the data, we consider maintaining local pairwise patterns using Locality Preserving Projections (LPP) (He & Niyogi, 2004). LPP will preserve the closeness and structure of nearby data points after the data has been projected into a lower dimensional space. Given a pairwise similarity matrix of the additional data, $\mathbf{S}$, LPP preserves the local relationships and maximizes the smoothness of the manifold of the data in the embedding space by minimizing (He & Niyogi, 2004),

$$\mathcal{J}_{LPP} = \sum_{i=1}^{n_i} \sum_{j=1}^{n_i} s_{ij} ||\mathbf{W}^T \mathbf{x}_i - \mathbf{W}^T \mathbf{x}_j||_2^2. \tag{3}$$

Our goal is to learn a dimensionally reduced representation of the longitudinal data that preserves both the local and global patterns in the data by combining the PCA and LPP objectives. We determine a projection by solving,

$$\mathcal{J}_{combined} = \mathcal{J}_{LPP} - \alpha \mathcal{J}_{PCA}, \qquad s.t. \mathbf{W}^T \mathbf{W} = \mathbf{I}, \tag{4}$$

where $0 \leq \alpha \leq 1$ is the tradeoff hyperparameter between the two projections. Empirically, the optimal $\alpha$ is difficult and time-consuming to calculate. To avoid calculating alpha, we present a quotient formulation that is equivalent in balancing between the objectives while inherently calculating $\alpha$,

$$\min_{\mathbf{W}} \frac{\sum_{i=1}^{n_i} \sum_{j=1}^{n_i} s_{ij} ||\mathbf{W}^T \mathbf{x}_i - \mathbf{W}^T \mathbf{x}_j|_2^2}{\sum_{i=1}^{n_i} ||\mathbf{W}^T \mathbf{x}_i||_2^2}, \qquad s.t. \mathbf{W}^T \mathbf{W} = \mathbf{I}. \tag{5}$$

Solving the objective $\frac{\mathcal{J}_{LPP}}{\mathcal{J}_{LPP}}$ results in a projection that benefits from both PCA and LPP, while preserving their constraints.

To support the constraint for this objective we rewrite the problem such that the constraint must hold true. We can verify that if $\mathbf{W}^T \mathbf{W} = \mathbf{I}$ the following equation holds,

$$\left\| \mathbf{b}_i - \mathbf{W} \mathbf{W}^T \mathbf{b}_i \right\|_2^2 = \|\mathbf{b}_i\|_2^2 - \left\| \mathbf{W}^T \mathbf{b}_i \right\|_2^2, \tag{6}$$

by which we can rewrite the problem as,

$$\min_{\mathbf{W}} \frac{\sum_{i=1}^{n_i} \sum_{j=1}^{n_i} s_{ij} ||\mathbf{W}^T \mathbf{x}_i - \mathbf{W}^T \mathbf{x}_j||_2^2}{\sum_{j=1}^{n_i} ||\mathbf{x}_j||_2^2 - |\mathbf{x}_j - \mathbf{W} \mathbf{W}^T \mathbf{x}_j||_2^2}, \qquad s.t. \mathbf{W}^T \mathbf{W} = \mathbf{I}. \tag{7}$$

Additionally, since the squared norms are susceptible to outliers, we convert all norms to $p$-order where $0 \leq p \leq 2$ to allow for tuned outlier sensitivity. This results in the problem as,

$$\min_{\mathbf{W}} \frac{\sum_{i=1}^{n_i} \sum_{j=1}^{n_i} s_{ij} ||\mathbf{W}^T \mathbf{x}_i - \mathbf{W}^T \mathbf{x}_j||_2^p}{\sum_{j=1}^{n_i} ||\mathbf{x}_j||_2^p - ||\mathbf{x}_j - \mathbf{W} \mathbf{W}^T \mathbf{x}_j||_2^p}, \qquad s.t. \mathbf{W}^T \mathbf{W} = \mathbf{I}. \tag{8}$$

## 3   Algorithm to convert ratio minimization to difference minimization

Before solving the problem in Equation 8, we first consider the more general problem,

$$\min_{v \in \mathcal{C}} \frac{f(v)}{g(v)}, \quad \text{where } g(v) \geq 0 \ (\forall \ v \in \mathcal{C}) \tag{9}$$

We require that the problem presented in Equation 9 be lower bounded, which is true for Equation 8. The algorithm to solve Equation 9 is presented in Algorithm 1. In this section we present theorems that prove that the algorithm converges to the global optimal solution, and at a quadratic convergence rate. The detailed proof of these theorems is provided in the appendix.

---

**Algorithm 1:** Algorithm to solve Equation 9.

1 Initialize $v \in \mathcal{C}$;
2 **while** *not converge* **do**
3      Calculate $\lambda = \frac{f(v)}{g(v)}$;
4      Update $v$ by solving the following problem:

$$v = \arg\min_{v \in \mathcal{C}} f(v) - \lambda g(v) \tag{10}$$

---

**Theorem 1** *(Wang et al., 2014) Algorithm 1 decreases the objective value of problem 9 in each iteration if the updated $v$ in Equation 10 satisfies $f(v) - \lambda g(v) \geq 0$.*

**Theorem 2** *(Wang et al., 2014) Algorithm 1's convergence rate is quadratic.*

**Theorem 3** *If the updated $v$ in Equation 10 of Algorithm 1 is a stationary point of problem 10, the converged solution in Algorithm 1 is a stationary point of problem 9.*

**Theorem 4** *Algorithm 1 decreases the objective value of the problem outlined in Equation 9 in each iteration if the updated value of $v$ in Equation 10 satisfies $f(v) - \lambda g(v) \leq 0$.*

**Theorem 5** *Algorithm 1's convergence rate is quadratic.*

Theorem 5 indicates that Algorithm 1 converges very fast to either a global minimum (if convex) or a local minimum (otherwise), *i.e.*, the difference between the current objective value and the optimal value is smaller than $\frac{1}{c^{2^t}}$, where $c > 1$ is a constant, at the $t$-th iteration.

Our formulation of the problem in Equation 8 fits into this general problem since the denominator is non-negative. The resulting formulation becomes,

$$\min_{\mathbf{W}} \sum_{i=1}^{n_i} \sum_{j=1}^{n_i} s_{ij} ||\mathbf{W}^T \mathbf{x}_i - \mathbf{W}^T \mathbf{x}_j||_2^p - \lambda(\sum_{j=1}^{n_i} ||\mathbf{x}_j||_2^p - ||\mathbf{x}_j - \mathbf{W}\mathbf{W}^T \mathbf{x}_j||_2^p), \tag{11}$$
$$s.t. \ \mathbf{W}^T \mathbf{W} = \mathbf{I}.$$

Here, we can remove the $\sum_{j=1}^{n_i} ||\mathbf{x}_j||_2^2$ term since it is a constant for any given data set for a given $\lambda$ in any iteration, resulting in the problem being,

$$\min_{\mathbf{W}} \sum_{i=1}^{n_i} \sum_{j=1}^{n_i} s_{ij} ||\mathbf{W}^T \mathbf{x}_i - \mathbf{W}^T \mathbf{x}_j||_2^p + \lambda \sum_{j=1}^{n_i} ||\mathbf{x}_j - \mathbf{W}\mathbf{W}^T \mathbf{x}_j||_2^p \qquad s.t. \mathbf{W}^T \mathbf{W} = \mathbf{I}. \tag{12}$$

## 4    SMOOTHED ITERATIVE REWEIGHTED METHOD

The problem in Equation 8 is a non-smooth objective and is difficult to efficiently solve in general. In this section, we introduce the *smoothed iterative reweighted method* to solve this optimization problem.

First, let us consider the general problem,

$$\min_{x \in \mathcal{C}} f(x) + \sum_i \mathbf{tr}\left((g_i^T(x)g_i(x))^{\frac{p}{2}}\right) . \tag{13}$$

When $g_i(x)$ is a vector output function, $\mathbf{tr}\left((g_i^T(x)g_i(x))^{\frac{p}{2}}\right)$ becomes,

$$\mathbf{tr}\left((g_i^T(x)g_i(x))^{\frac{p}{2}}\right) = ||g_i(x)||_2^p . \tag{14}$$

Equation 13 is non-smooth, so we turn to solve the following smooth problem

$$\min_{x \in C} f(x) + \sum_i \mathbf{tr}((g_i^T(x)g_i(x) + \delta I)^{\frac{p}{2}}) , \tag{15}$$

where $\delta > 0$ is a small constant. When $\delta \to 0$, Equation 15 is reduced to Equation 13 since,

$$\lim_{\delta \to 0} \mathbf{tr}((g_i^T(x)g_i(x) + \delta I)^{\frac{p}{2}}) = ||g_i(x)||_2^p . \tag{16}$$

We derive Algorithm 2 to solve the problem in Equation 15 with the detailed derivation provided in the appendix. The convergence of Algorithm 2 is guaranteed with the proof in the appendix.

---

**Algorithm 2:** Algorithm to solve Equation 15.

1 Initialize $x \in \mathcal{C}$ ;
2 **while** *not converge* **do**
3     1. For each $i$, calculate $D_i = \frac{p}{2}(g_i^T(x)g_i(x) + \delta I)^{\frac{p-2}{2}}$ ;
4     2. Update $x$ by solving the problem $\min_{x \in \mathcal{C}} f(x) + \sum_i \mathbf{tr}(g_i^T(x)g_i(x)D_i)$ ;
**Output:** $x$.

---

We note that the iterative reweighted method introduced in (Candes et al., 2008; Nie et al., 2010) solves the non-smooth $\ell_1$-norm or $\ell_{2,1}$-norm minimization problems. However, the method in (Candes et al., 2008; Nie et al., 2010) does not explicitly use the smoothness constant (*i.e.*, $\delta$ in Equation 15). Without the smoothness term, the algorithm is impacted by the singularity problem due to inverted matrices that divide 0s, which routinely lead to inferior learning performance. A smoothness term was informally used in (Nie et al., 2010; Wang et al., 2013; Nie et al., 2013) to improve numerical stability. However, the authors did not include it in their proof of the convergence of the algorithm. As a theoretical contribution of this paper, we formally introduce the smoothness term (*i.e.*, $\delta I$ in Equation 15) in our algorithm and theoretically prove the algorithm's convergence in which the smoothness term leads to more stable solutions. We refer to Algorithm 2 as the *Smoothed Iterative Reweighted Method*. We can represent our problem following this general problem as,

$$\min_{\mathbf{W}} \sum_{i=1}^{n_i} \sum_{j=1}^{n_i} \gamma_j s_{ij} ||\mathbf{W}^T \mathbf{x}_i - \mathbf{W}^T \mathbf{x}_j||_2^2 + \lambda(\sum_{j=1}^{n_i} \theta_j ||\mathbf{x}_j - \mathbf{W}\mathbf{W}^T \mathbf{x}_j||_2^2), \quad s.t. \mathbf{W}^T \mathbf{W} = \mathbf{I}, \tag{17}$$

where $\lambda = \dfrac{\sum_{i=1}^{n_i} \sum_{j=1}^{n_i} s_{ij} ||\mathbf{W}^T \mathbf{x}_i - \mathbf{W}^T \mathbf{x}_j||_2^p}{\sum_{j=1}^{n_i} ||\mathbf{x}_j||_2^p - ||\mathbf{x}_j - \mathbf{W}\mathbf{W}^T \mathbf{x}_j||_2^p}$, (18)

$$\gamma_j = \frac{p}{2}(\sum_{i=1}^{n_i} s_{ij} ||\mathbf{W}^T \mathbf{x}_i - \mathbf{W}^T \mathbf{x}_j||_2^2 + \delta_\gamma)^{\frac{p-2}{2}} \tag{19}$$

$$\theta_j = \frac{p}{2}(||\mathbf{x}_j - \mathbf{W}\mathbf{W}^T \mathbf{x}_j||_2^2 + \delta_\theta)^{\frac{p-2}{2}} \tag{20}$$

We can simplify this problem's presentation by introducing $\tilde{s}_{ij} = \gamma_j s_{ij}$ and $\tilde{\theta}_j = \lambda \theta_j$ resulting in,

$$\min_{\mathbf{W}} \sum_{i=1}^{n_i} \sum_{j=1}^{n_i} \tilde{s}_{ij} ||\mathbf{W}^T \mathbf{x}_i - \mathbf{W}^T \mathbf{x}_j||_2^2 + \sum_{j=1}^{n_i} \tilde{\theta}_j ||\mathbf{x}_j - \mathbf{W}\mathbf{W}^T \mathbf{x}_j||_2^2, \quad s.t. \mathbf{W}^T \mathbf{W} = \mathbf{I}. \tag{21}$$

To solve this problem, we follow the iterative algorithm outlined in Algorithm 3. To determine a solution to the two-order problem, we rewrite the objective in matrix form. We use $\mathbf{\Gamma}$ and $\mathbf{\Theta}$ to represent diagonal matrixes in $\mathbb{R}^{n_i \times n_i}$ with each value representing $\gamma_j$ and $\theta_j$ respectively. Therefore, $\tilde{\mathbf{S}} = \mathbf{\Gamma}\mathbf{S}$ and $\tilde{\mathbf{\Theta}} = \mathbf{\Theta}\lambda$. We also introduce the Laplacian matrix, $\mathbf{L} = \mathbf{D} - \tilde{\mathbf{S}}$ where $\mathbf{D}$ is the diagonal degree matrix with $d_{ii} = \sum_{j=1}^{n_i} \tilde{s}_{ij}$. We rewrite the problem as,

$$\min_{\mathbf{W}} \mathbf{tr}(\mathbf{W}^T \mathbf{X} \mathbf{L} \mathbf{X}^T \mathbf{W}) + \mathbf{tr}((\mathbf{X} - \mathbf{W}\mathbf{W}^T \mathbf{X})\tilde{\mathbf{\Theta}}(\mathbf{X}^T - \mathbf{X}^T \mathbf{W}\mathbf{W}^T)) \tag{22}$$

$$= \mathbf{tr}(\mathbf{W}^T(\mathbf{X}\mathbf{L}\mathbf{X}^T - \mathbf{X}\tilde{\mathbf{\Theta}}\mathbf{X}^T)\mathbf{W}), \quad s.t. \mathbf{W}^T \mathbf{W} = \mathbf{I}. \tag{23}$$

We directly solve the problem using the Rayleigh quotient (Hom & Johnson, 1985) resulting in $\mathbf{W}$ being the $r$ smallest eigenvectors of $\mathbf{X}\mathbf{L}\mathbf{X}^T - \mathbf{X}\tilde{\mathbf{\Theta}}\mathbf{X}^T$. Our solution algorithm is outlined in Algorithm 3.

---

**Algorithm 3:** Algorithm to solve problem outlined in Equation 8

---

**1** Initialize $\mathbf{W}$
**2** **while** *not converge* **do**
**3**     Update $\lambda$ using Equation 18
**4**     **while** *not converge* **do**
**5**         Update $\gamma_j$ using Equation 19
**6**         Update $\tilde{\theta}_j$ using Equation 20
**7**         Update $\mathbf{W}$ as the smallest $r$ eigenvectors of $\mathbf{XLX}^T - \mathbf{X}\tilde{\Theta}\mathbf{X}^T$
**8**     **end**
**9** **end**

---

## 5   EXPERIMENTS & RESULTS

To empirically support the value of our method, we conduct several experiments. First, we analyze the convergence of our solution algorithm, ensuring that it in fact converges as theoretically proven. Second, we perform a parameter study and conduct a classification experiment using our method at varying values of outlier sensitivity, $p$. We then compare the performance of the best enriched representation after hyperparameter tuning with the baseline representation at predicting patients' diagnoses using classical classification methods. Finally, we investigate the interpretability of our method by checking the coefficients it allocates to each feature when enriching the data.

For our experiments, we used brain imaging scans from the Alzheimer's Disease Neuroimaging Initiative (ADNI) of 544 patients. The patients in our experiments all had a baseline scan and between 3 and 5 additional scans taken at various time-points of at least 6 month increments: 6 months, 12 months, 18 months, 24 months, and 36 months. Each patient's diagnosis belongs to one of the following classes; Alzheimers Disease (AD) - 92 patients, Mild Cognitive Impairment (MCI) - 205 patients, and Healthy Control (HC) - 177 patients. We performed voxel-based morphometry (VBM) on the MRI data by following (Risacher et al., 2010) and extracted mean modulated gray matter measures for 90 target regions of interest (ROI). These measures are adjusted for the baseline intracranial volume using regression weights derived from the HC participants at baseline. Our selected value for the dimensionality of the enriched representation was set at $r = 3$ for all experiments.

### 5.1   ALGORITHM CONVERGENCE ANALYSIS

We monitor the objective function's value over many iterations for several subjects to monitor its convergence. The value of the objective function over time is shown in Figure 2. Each line in the figure represents a different patient, showing different rates of convergence depending on the individual's data. The objective generally converges at a fast rate of under 10 iterations.

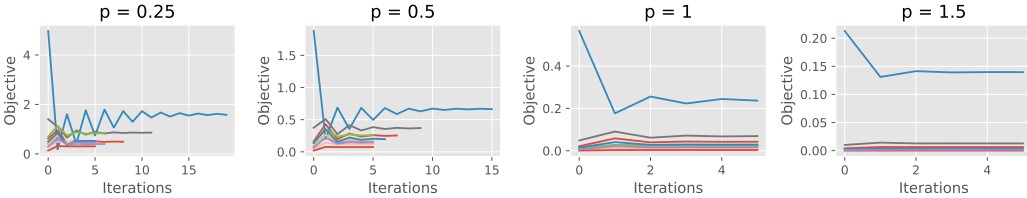

Figure 2: Algorithm convergence for 10 example subjects across different values of $p$.

### 5.2   PARAMETER STUDY

A powerful attribute of our method's approach is the ability to tune outlier sensitivity, $p$. Here we monitor the effect of changing the value of $p$ on classification performance. We use both accuracy and weighted F-1 scores to measure the performance of our classification experiments that we conduct on a 10-fold cross validation test, taking into consideration the standard deviation of our

metrics. We run the experiment with Decision Trees, Gaussian Naive Bayes, Multi-Layer Perceptrons (MLP), Support Vector Machines (SVM), and $k$-Nearest Neighbors ($k$NN) models following a hyperparameter search. The results for accuracy and weighted F-1 scores are shown in Table 1 and 2 respectively where we present the mean score and 95% confidence interval. Bolded results represent the best results after taking into consideration statistical $p$-values of a pairwise T-test when compared to the baseline results. Methods that do not have a bold value mean that the results were not significant enough for us to confidently state an improved diagnosis performance. The statistical $p$-values are provided in tables in the appendix.

The best value for outlier sensitivity $p$ can often vary and in this case the most common superior model utilizes $p = 0.25$. This shows the value in modifying our distance choices from a simple 2-order distance. The outlier sensitivity tuning allows us to find a better result for our model that customizes the weights of outliers versus other points. We believe that models of different applications should account for outliers differently and therefore there is value in providing this model customization. Our results provide supporting evidence to our case.

| View | | Decision Tree | Naive Bayes | MLP | SVM | $k$-NN |
|---|---|---|---|---|---|---|
| Baseline | | $0.455 \pm 0.107$ | $0.397 \pm 0.108$ | $0.496 \pm 0.113$ | $0.542 \pm 0.165$ | $0.484 \pm 0.124$ |
| | p=0.25 | $\mathbf{0.552 \pm 0.113}$ | $\mathbf{0.588 \pm 0.0954}$ | $0.523 \pm 0.0457$ | $0.518 \pm 0.0461$ | $0.549 \pm 0.108$ |
| | p=0.5 | $0.477 \pm 0.129$ | $0.462 \pm 0.0581$ | $0.514 \pm 0.0091$ | $0.514 \pm 0.0091$ | $0.48 \pm 0.0978$ |
| Enriched | p=1 | $0.442 \pm 0.0766$ | $0.495 \pm 0.0324$ | $0.514 \pm 0.0091$ | $0.514 \pm 0.0091$ | $0.489 \pm 0.0947$ |
| | p=1.5 | $0.46 \pm 0.15$ | $0.504 \pm 0.0245$ | $0.514 \pm 0.0091$ | $0.514 \pm 0.0091$ | $0.491 \pm 0.0924$ |
| | p=2 | $0.527 \pm 0.0995$ | $0.498 \pm 0.0657$ | $0.532 \pm 0.0954$ | $0.59 \pm 0.12$ | $\mathbf{0.574 \pm 0.0933}$ |

Table 1: Accuracy scores for various values of p across different classification methods.

| View | | Decision Tree | Naive Bayes | MLP | SVM | $k$-NN |
|---|---|---|---|---|---|---|
| Baseline | | $0.453 \pm 0.102$ | $0.372 \pm 0.099$ | $0.331 \pm 0.113$ | $0.48 \pm 0.211$ | $0.444 \pm 0.12$ |
| | p=0.25 | $\mathbf{0.55 \pm 0.0615}$ | $\mathbf{0.539 \pm 0.0831}$ | $0.389 \pm 0.129$ | $0.383 \pm 0.0658$ | $0.532 \pm 0.111$ |
| | p=0.5 | $0.474 \pm 0.128$ | $0.417 \pm 0.0687$ | $0.35 \pm 0.0103$ | $0.381 \pm 0.0465$ | $0.437 \pm 0.118$ |
| Enriched | p=1 | $0.433 \pm 0.101$ | $0.354 \pm 0.0458$ | $0.35 \pm 0.0103$ | $0.35 \pm 0.0103$ | $0.45 \pm 0.0973$ |
| | p=1.5 | $0.455 \pm 0.158$ | $0.353 \pm 0.0234$ | $0.384 \pm 0.101$ | $0.35 \pm 0.0103$ | $0.458 \pm 0.126$ |
| | p=2 | $0.518 \pm 0.0882$ | $0.367 \pm 0.0634$ | $0.439 \pm 0.221$ | $0.569 \pm 0.114$ | $\mathbf{0.564 \pm 0.0894}$ |

Table 2: Weighted F-1 scores for various values of p across different classification methods.

## 5.3 SUCCESSFUL DIAGNOSIS RATE COMPARISON

Following our parameter study, we compare the performance of the classification models on the baseline data to the performance of the models using the data longitudinally enriched with our learned projections. We also present the statistical $p$-value of the T-test for paired samples given that we used the same folds across experiments. The resulting scores presented in Table 1 and 2 also compare our method's performance to the baseline representation. Our method significantly outperforms the baseline representation when used with Decision Trees, Naive Bayes, and $k$-NN, showing the value added from the available additional data for patients and the enriched representation.

## 5.4 MODEL INTERPRETABILITY

A key feature in our method's approach is interpretability in terms of the key biomarkers that differentiate the various patient scans. Including PCA in the projection allows us to use the learned coefficients of the projection, $\mathbf{W}$, to determine the importance of key biomarkers. Since we have a projection per patient, we can determine the importance of each biomarker for that patient in particular. We conduct a brief summary of all the patients' projections to determine the most important biomarkers across patients. We tallied the 10 most frequently appearing regions of interest from each patient's top 10 projection coefficients and normalized the results to determine the relative weights of each coefficient. These results presented the key unsupervised biomarkers that are learned from the data. These biomarkers are key to differentiating global and local patterns in the data across subjects in the ADNI without any supervision regarding their medical condition. We also repeated

Figure 3: Key biomarkers learned from unsupervised patients' projections.

Figure 4: Key biomarkers learned from AD classification.

this analysis by using the best performing decision tree model's feature weights multiplied by each patient's projection coefficient to determine the weights with respect to classifying AD. This second approach provides a supervised approach to relating key biomarkers with AD specifically. Results from both the unsupervised and supervised analyses are shown in Figure 3 and 4 respectively.

We investigated the top biomarkers in our enriched feature representation and determine their relationship to AD. The following biomarkers were found to have the highest relevance to the enriched representation when weighted by the best performing decision tree model's feature importances; putamen, thalamus, amygdala, hippocampus, caudate nucleus, and Heschl's gyrus. We find that our results are generally consistent with medical research on AD.

The putamen and thalamus are key biomarkers in diffentiating between patients (Bollen et al., 2008). The amygdala is found to be a key region affecting AD in (Poulin et al., 2011; Laakso et al., 1995). The left and right hippocampal regions are found to be in the top selected biomarkers by our model with support from (Laakso et al., 1995). The caudate nucleus has some supporting evidence from (Jiji et al., 2013) where it was found to exhibit heavy correlation with AD. More recently, (Persson et al., 2017) found more supporting evidence of the relationship.

The extensive support by medical research of key biomarkers our method has determined elicit confidence in ourapproach. Our method shows that it can encapsulate key information from various brain regions over time and distill that in a consistent, small dimensional representation. The following biomarkers were found to have the highest relevance in patient differentiation by our unsupervised method without weighting for diagnosis prediction; putamen, thalamus, amygdala, parahippocampal gyrus, caudate nucleus, olfactory sulcus, fusiform gyrus, insular cortex, middle temporal gyrus, and lingual gyrus. The key biomarkers for AD are still included, however the representation encompasses more than just the focus on AD. In addition to memory retrieval and processing, the biomarkers represent other brain functions such as vision, olfaction, recognition, and cognition. This allows us to obtain a representation that isn't only focused on AD but can represent a patient holistically across time-points regardless of the task to be performed.

## 6  CONCLUSION

Longitudinal data with inconsistently available sample data can cause difficulty with learning consistent data representations. This is evident in medical studies such as the Alzheimers Disease Neuroimaging Initiative. Our proposed method calculates a patient-specific projection that allows different patients to have varying additional data and produces a consistent-length representation for all patients by enriching the shared baseline with their additional data. Our results show that our enriched representation surpasses the performance of the baseline representation at predicting patient diagnoses.

A limitation of our method is that the dimensionality of the enriched representation is limited by the minimum number of additional data a patient has, *i.e.* $r < min_{\forall i} n_i$. This limitation isn't often impactful due to large amounts of available data for longitudinal problems. In addition to the superior performance of predicting diagnoses, our method provides both unsupervised and supervised interpretability. The resulting coefficients of key regions of interest of the brain are consistent with medical research studying the brain and AD's effects. Our method can achieve superior performance by enriching consistent data with additionally available data, thereby more accurately predicting AD and potentially improving or extending many patients' lives.

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

## A  APPENDIX

Before deriving the algorithm to optimize the problem in Equation 15, we need the following proofs and lemmas.

**Proof A.1** *In each iteration of Algorithm 1, we represent the updated $v$ in Equation 10 as $\tilde{v}$ which satisfies $f(\tilde{v}) - \lambda g(\tilde{v}) \leq f(v) - \lambda g(v)$. Thus $f(\tilde{v}) - \lambda g(\tilde{v}) \leq 0$, which indicates that $\frac{f(\tilde{v})}{g(\tilde{v})} \leq \lambda = \frac{f(v)}{g(v)}$ because $g(\tilde{v}) \geq 0$.*

**Proof A.2** *The Lagrangian function of the problem in Equation 10 is*

$$\mathcal{L}_1(v, \alpha_1, \beta_1) = f(v) - \lambda g(v) + \alpha_1^T p(v) + \beta_1^T q(v). \tag{24}$$

*We denote the converged solution in Algorithm 1 as $v^*$. If $v^*$ is a stationary point of problem 10, we have*

$$f'(v^*) - \frac{f(v^*)}{g(v^*)} g'(v^*) + \alpha_1^T p'(v^*) + \beta_1^T q'(v^*) = 0, \tag{25}$$

*which can be written as*

$$\frac{g(v^*)f'(v^*) - f(v^*)g'(v^*)}{g^2(v^*)} + \frac{\alpha_1^T}{g(v^*)} p'(v^*) + \frac{\beta_1^T}{g(v^*)} q'(v^*) = 0.$$

*Let $\alpha_2 = \frac{\alpha_1}{g(v^*)}$ and $\beta_2 = \frac{\beta_1}{g(v^*)}$, then we have*

$$\left( \frac{f(v)}{g(v)} \right)' \Bigg|_{v = v^*} + \alpha_2^T p'(v^*) + \beta_2^T q'(v^*) = 0. \tag{26}$$

*Note that $\alpha_1 \geq 0$ and $g(v^*) \geq 0$, so $\alpha_2 \geq 0$. Therefore, $v^*$ is a stationary point to the Lagrangian function of problem 9 as follows:*

$$\mathcal{L}_2(v, \alpha_2, \beta_2) = \frac{f(v)}{g(v)} + \alpha_2^T p(v) + \beta_2^T q(v), \tag{27}$$

*which completes the proof.*

**Proof A.3** *We define a function $h(\lambda)$ as,*

$$h(\lambda) = \min_{v \in \mathcal{C}} \ f(v) - \lambda g(v). \tag{28}$$

*According to Algorithm 1, the converged $\lambda^*$ is the root of $h(\lambda)$, that is, $h(\lambda^*) = 0$. In each iteration of Algorithm 1, we represent the updated $v$ in Equation 10 as $\tilde{v}$. According to line 4, $h(\lambda) = f(\tilde{v}) - \lambda g(\tilde{v})$. Thus $h'(\lambda) = -g(\tilde{v})$.*

*Using Newton's method, the updated solution should be $\tilde{\lambda} = \lambda - \frac{h(\lambda)}{h'(\lambda)} = \lambda - \frac{f(\tilde{v}) - \lambda g(\tilde{v})}{-g(\tilde{v})} = \frac{f(\tilde{v})}{g(\tilde{v})}$, which is equivalent to step 1 in the next iteration of Algorithm 1. Algorithm 1 is a Newton's method solution to find the root of $h(\lambda)$, and so its convergence rate is quadratic.*

Based on the chain rule, we have:

**Lemma 1** *Suppose $g(x)$ is a matrix output function, $h(x)$ is a scalar output function, and $x$ is a scalar, vector or matrix variable, then*

$$\frac{\partial h(g(x))}{\partial x} = \frac{\sum_{i,j} \frac{\partial h(g(x))}{\partial g_{ij}(x)} \partial g_{ij}(x)}{\partial x} = \left( \frac{\partial h(g(x))}{\partial g(x)} \right)^T \frac{\partial g(x)}{\partial x} . \tag{29}$$

According to the chain rule in Lemma 1, we can derive the following two lemmas:

**Lemma 2** *Suppose $g(x)$ is a scalar, vector or matrix output function, and $x$ is a scalar, vector, or matrix variable, then*

$$\frac{\partial \boldsymbol{tr}((g^T(x)g(x) + \delta I)^{\frac{p}{2}})}{\partial x} = p \, (g^T(x)g(x) + \delta I)^{\frac{p-2}{2}} g^T(x) \frac{\partial g(x)}{\partial x} . \tag{30}$$

**Lemma 3** *Suppose $g(x)$ is a scalar, vector or matrix output function, $x$ is a scalar, vector or matrix variable, and $D$ is a constant that is symmetrical if it is a matrix, then*

$$\frac{\partial \textbf{\textit{tr}}(g^T(x)g(x)D)}{\partial x} = 2Dg^T(x)\frac{\partial g(x)}{\partial x} \quad. \tag{31}$$

Now we derive the algorithm to optimize the problem in Equation 15. The Lagrangian function of the problem in Equation 15 is

$$\mathcal{L}(x, \lambda) = f(x) + \sum_i \textbf{tr}((g_i^T(x)g_i(x) + \delta I)^{\frac{p}{2}}) - r(x, \lambda) \quad, \tag{32}$$

where $r(x, \lambda)$ is a Lagrangian term for the constraint $x \in \mathcal{C}$. By setting the derivative of Equation 32) wrt $x$ to zero, we have

$$\frac{\partial L(x, \lambda)}{\partial x} = f'(x) + \sum_i \frac{\partial \textbf{tr}((g_i^T(x)g_i(x) + \delta I)^{\frac{p}{2}})}{\partial x} - \frac{\partial r(x, \lambda)}{\partial x} = 0 \quad. \tag{33}$$

According to Lemma 2, Equation 33 can be rewritten as

$$f'(x) + \sum_i p(g_i^T(x)g_i(x) + \delta I)^{\frac{p-2}{2}} g_i^T(x)\frac{\partial g_i(x)}{\partial x} - \frac{\partial r(x, \lambda)}{\partial x} = 0 \quad. \tag{34}$$

If we can find a solution, $x$, that satisfies Equation 34, we can find a local or global optimal solution to the problem in Equation 15 according to the Karush-Kuhn-Tucker (KKT) conditions (Boyd & Vandenberghe, 2004). Following (Candes et al., 2008; Nie et al., 2010), we propose an iterative algorithm to find a solution to Equation 34 using the observation that $D_i = \frac{p}{2}(g_i^T(x)g_i(x) + \delta I)^{\frac{p-2}{2}}$, is a constant. Equation 34 is reduced to:

$$f'(x) + \sum_i 2D_i g_i^T(x)\frac{\partial g_i(x)}{\partial x} - \frac{\partial r(x, \lambda)}{\partial x} = 0 \quad, \tag{35}$$

which is equivalent to solving

$$\min_{x \in \mathcal{C}} f(x) + \sum_i \textbf{tr}(g_i^T(x)g_i(x)D_i) \quad. \tag{36}$$

Based on these observations, we can first guess a solution to Equation 15, $x$. Then we calculate $D_i$ using the current value of $x$ and update $x$ by the optimal solution of the problem in Equation 36 by calculating $D_i$. We iteratively perform this procedure until it converges, which is summarized in Algorithm 2.

The convergence of Algorithm 2 is guaranteed by the following theorem. (The proof of Theorem 6 is supplied in the appendix due to space limitations.)

**Theorem 6** *Algorithm 2 will monotonically decrease the objective of the problem in equation 15 in each iteration until it converges.*

During convergence, Equation 34 will hold, thus the KKT conditions of the problem in Equation 15 are satisfied. Therefore, Algorithm 2 will converge to a local optimal solution to the problem in Equation 15. If the problem in Equation 15 is convex, Algorithm 2 will converge to a global optimal solution.

PROOF OF THEOREM 6

Before proving the convergence of Algorithm 2, we first introduce several lemmas:

**Lemma 4** *For any $\sigma > 0$, the following inequality holds when $0 < p \leq 2$:*

$$\frac{p}{2}\sigma - \sigma^{\frac{p}{2}} + \frac{2-p}{2} \geq 0 \quad. \tag{37}$$

**Proof.** Denoting $f(\sigma) = p\sigma - 2\sigma^{\frac{p}{2}} + 2 - p$, we have the following derivatives:

$$f'(\sigma) = p(1 - \sigma^{\frac{p-2}{2}}), \quad \text{and} \quad f''(\sigma) = \frac{p(2-p)}{2}\sigma_i^{\frac{p-4}{2}}.$$

When $\sigma > 0$ and $0 < p \leq 2$, $f''(\sigma) \geq 0$ and $\sigma = 1$ is the only point that $f'(\sigma) = 0$. Note that $f(1) = 0$, thus when $\sigma > 0$ and $0 < p \leq 2$, then $f(\sigma) \geq 0$, which indicates Inequality 37 holds true. $\square$

**Lemma 5** *(Ruhe, 1970) For any positive definite matrices $\tilde{M}, M$ with the same size, suppose the eigen-decompositions $\tilde{M} = U\Sigma U^T$, $M = V\Lambda V^T$, where the eigenvalues in $\Sigma$ are in increasing order and the eigenvalues in $\Lambda$ are in decreasing order. Then the following inequality holds when $0 < p \leq 2$:*

$$\boldsymbol{tr}(\tilde{M}M) \geq \boldsymbol{tr}(\Sigma\Lambda) \ . \tag{38}$$

**Lemma 6** *For any positive definite matrices $\tilde{M}, M$ with the same size, the following inequality holds when $0 < p \leq 2$:*

$$\boldsymbol{tr}(\tilde{M}^{\frac{p}{2}}) - \frac{p}{2}\boldsymbol{tr}(\tilde{M}M^{\frac{p-2}{2}}) \leq \boldsymbol{tr}(M^{\frac{p}{2}}) - \frac{p}{2}\boldsymbol{tr}(MM^{\frac{p-2}{2}}). \tag{39}$$

**Proof.** For any $\sigma > 0$, $\lambda > 0$ and $0 < p \leq 2$, according to Lemma 4 we have $\frac{p}{2}(\frac{\sigma}{\lambda}) - (\frac{\sigma}{\lambda})^{\frac{p}{2}} + \frac{2-p}{2} \geq 0$, which indicates

$$\frac{p}{2}\sigma\lambda^{\frac{p-2}{2}} - \sigma^{\frac{p}{2}} + \frac{2-p}{2}\lambda^{\frac{p}{2}} \geq 0 \ . \tag{40}$$

Suppose the eigen-decomposition $\tilde{M} = U\Sigma U^T$, $M = V\Lambda V^T$, where the eigenvalues in $\Sigma$ is in increasing order and the eigenvalues in $\Lambda$ is in decreasing order. Then according to Inequality 40, we have

$$\frac{p}{2}\boldsymbol{tr}(\Sigma\Lambda^{\frac{p-2}{2}}) - \boldsymbol{tr}(\Sigma^{\frac{p}{2}}) + \frac{2-p}{2}\boldsymbol{tr}(\Lambda^{\frac{p}{2}}) \geq 0 \ , \tag{41}$$

and according to Lemma 5 we have

$$\frac{p}{2}\boldsymbol{tr}(\tilde{M}M^{\frac{p-2}{2}}) - \frac{p}{2}\boldsymbol{tr}(\Sigma\Lambda^{\frac{p-2}{2}}) \geq 0 \ , \tag{42}$$

$$\frac{p}{2}\boldsymbol{tr}(\tilde{M}M^{\frac{p-2}{2}}) - \boldsymbol{tr}(\Sigma^{\frac{p}{2}}) + \frac{2-p}{2}\boldsymbol{tr}(\Lambda^{\frac{p}{2}}) \geq 0 \ . \tag{43}$$

Note that $\boldsymbol{tr}(\tilde{M}^{\frac{p}{2}}) = \boldsymbol{tr}(\Sigma^{\frac{p}{2}})$ and $\boldsymbol{tr}(M^{\frac{p}{2}}) = \boldsymbol{tr}(\Lambda^{\frac{p}{2}})$, so we have

$$\frac{p}{2}\boldsymbol{tr}(\tilde{M}M^{\frac{p-2}{2}}) - \boldsymbol{tr}(\tilde{M}^{\frac{p}{2}}) + \frac{2-p}{2}\boldsymbol{tr}(M^{\frac{p}{2}}) \geq 0$$
$$\Rightarrow \boldsymbol{tr}(\tilde{M}^{\frac{p}{2}}) - \frac{p}{2}\boldsymbol{tr}(\tilde{M}M^{\frac{p-2}{2}}) \leq \frac{2-p}{2}\boldsymbol{tr}(M^{\frac{p}{2}})$$
$$\Rightarrow \boldsymbol{tr}(\tilde{M}^{\frac{p}{2}}) - \frac{p}{2}\boldsymbol{tr}(\tilde{M}M^{\frac{p-2}{2}}) \leq \boldsymbol{tr}(M^{\frac{p}{2}}) - \frac{p}{2}\boldsymbol{tr}(MM^{\frac{p-2}{2}}) \ ,$$

which completes the proof. $\square$

**Lemma 7** *For any matrices $\tilde{A}, A$ with the same size and $\delta > 0$, the following inequality holds when $0 < p \leq 2$:*

$$\boldsymbol{tr}((\tilde{A}^T\tilde{A} + \delta I)^{\frac{p}{2}}) - \frac{p}{2}\boldsymbol{tr}(\tilde{A}^T\tilde{A}(A^TA + \delta I)^{\frac{p-2}{2}})$$
$$\leq \boldsymbol{tr}((A^TA + \delta I)^{\frac{p}{2}}) - \frac{p}{2}\boldsymbol{tr}(A^TA(A^TA + \delta I)^{\frac{p-2}{2}}) \ . \tag{44}$$

**Proof.** Note that $\tilde{A}^T\tilde{A} + \delta I$ and $A^TA + \delta I$ are positive definite matrices since $\delta > 0$. Then according to Lemma 6 we have

$$\boldsymbol{tr}((\tilde{A}^T\tilde{A} + \delta I)^{\frac{p}{2}}) - \frac{p}{2}\boldsymbol{tr}((\tilde{A}^T\tilde{A} + \delta I)(A^TA + \delta I)^{\frac{p-2}{2}})$$
$$\leq \boldsymbol{tr}((A^TA + \delta I)^{\frac{p}{2}}) - \frac{p}{2}\boldsymbol{tr}((A^TA + \delta I)(A^TA + \delta I)^{\frac{p-2}{2}}) \ , \tag{45}$$

which indicates that Inequality 44 holds true. $\square$

As a result, we can prove Theorem 6 as follows now.

In step 2 of Algorithm 2, we denote the updated $x$ as $\tilde{x}$. According to step 2, we know

$$f(\tilde{x}) + \sum_i \boldsymbol{tr}(g_i^T(\tilde{x})g_i(\tilde{x})D_i) \leq f(x) + \sum_i \boldsymbol{tr}(g_i^T(x)g_i(x)D_i) \ , \tag{46}$$

where the inequality holds when and only when the algorithm converges.

For each $i$, according to Lemma 7, we have

$$
\begin{aligned}
&\mathbf{tr}((g_i^T(\tilde{x})g_i(\tilde{x}) + \delta I)^{\frac{p}{2}}) \\
&- \tfrac{p}{2}\mathbf{tr}(g_i^T(\tilde{x})g_i(\tilde{x})(g_i^T(x)g_i(x) + \delta I)^{\frac{p-2}{2}}) \\
&\leq \mathbf{tr}((g_i^T(x)g_i(x) + \delta I)^{\frac{p}{2}}) \\
&- \tfrac{p}{2}\mathbf{tr}(g_i^T(x)g_i(x)(g_i^T(x)g_i(x) + \delta I)^{\frac{p-2}{2}}) \ .
\end{aligned}
\tag{47}
$$

Note that $D_i = \frac{p}{2}(g_i^T(x)g_i(x) + \delta I)^{\frac{p-2}{2}}$, so for each $i$ we have

$$
\begin{aligned}
&\mathbf{tr}((g_i^T(\tilde{x})g_i(\tilde{x}) + \delta I)^{\frac{p}{2}}) - \mathbf{tr}(g_i^T(\tilde{x})g_i(\tilde{x})D_i) \\
&\leq \mathbf{tr}((g_i^T(x)g_i(x) + \delta I)^{\frac{p}{2}}) - \mathbf{tr}(g_i^T(x)g_i(x)D_i) \ .
\end{aligned}
\tag{48}
$$

Then we have

$$
\begin{aligned}
&\sum_i \mathbf{tr}((g_i^T(\tilde{x})g_i(\tilde{x}) + \delta I)^{\frac{p}{2}}) - \sum_i \mathbf{tr}(g_i^T(\tilde{x})g_i(\tilde{x})D_i) \\
&\leq \sum_i \mathbf{tr}((g_i^T(x)g_i(x) + \delta I)^{\frac{p}{2}}) - \sum_i \mathbf{tr}(g_i^T(x)g_i(x)D_i) \ .
\end{aligned}
\tag{49}
$$

Summing Inequality 46 and Inequality 49 on both sides, we arrive at

$$
\begin{aligned}
f(\tilde{x}) + \sum_i \mathbf{tr}((g_i^T(\tilde{x})g_i(\tilde{x}) + \delta I)^{\frac{p}{2}}) \leq \\
f(x) + \sum_i \mathbf{tr}((g_i^T(x)g_i(x) + \delta I)^{\frac{p}{2}}) \ .
\end{aligned}
\tag{50}
$$

Note that Inequality 50 holds only when the algorithm converges. Thus the Algorithm 2 will monotonically decrease the objective of the problem in Equation 15 in each iteration until the algorithm converges. $\qquad\square$

In the convergence, the equality in Equation 34 will hold, thus the KKT condition of problem 15 is satisfied. Therefore, the Algorithm 2 will converge to a local optimum solution to the problem 15. If the problem 15 is convex, the Algorithm 2 will converge to a global optimum solution.

# DETAILED EXPERIMENTAL RESULTS

The following tables (Tables 3 & 4) include the same results as the tables in the original content with the addition of p-values to denote the significance of the results when compared to the Baseline method. The results in these tables should clarify the reasoning behind the winning bolded results as shown in the original content.

Table 3: Accuracy scores for various values of p across different classification methods.

| View | | Decision Tree | Naive Bayes | MLP | SVM | $k$-NN |
|---|---|---|---|---|---|---|
| Baseline | | $0.455 \pm 0.107, -$ | $0.397 \pm 0.108, -$ | $0.496 \pm 0.113, -$ | $0.542 \pm 0.165, -$ | $0.484 \pm 0.124, -$ |
| | p=0.25 | $\mathbf{0.552 \pm 0.113, 0.00966}$ | $\mathbf{0.588 \pm 0.0954, 2.92e-06}$ | $0.523 \pm 0.0457, 0.287$ | $0.518 \pm 0.0461, 0.391$ | $0.549 \pm 0.108, 0.0203$ |
| | p=0.5 | $0.477 \pm 0.129, 0.453$ | $0.462 \pm 0.0581, 0.00795$ | $0.514 \pm 0.0091, 0.343$ | $0.514 \pm 0.0091, 0.343$ | $0.48 \pm 0.0978, 0.895$ |
| Enriched | p=1 | $0.442 \pm 0.0766, 0.553$ | $0.495 \pm 0.0324, 0.000213$ | $0.514 \pm 0.0091, 0.343$ | $0.514 \pm 0.0091, 0.343$ | $0.489 \pm 0.0947, 0.837$ |
| | p=1.5 | $0.46 \pm 0.15, 0.811$ | $0.504 \pm 0.0245, 0.000312$ | $0.514 \pm 0.0091, 0.343$ | $0.514 \pm 0.0091, 0.343$ | $0.491 \pm 0.0924, 0.747$ |
| | p=2 | $0.527 \pm 0.0995, 0.0292$ | $0.498 \pm 0.0657, 0.000198$ | $0.532 \pm 0.0954, 0.144$ | $0.59 \pm 0.12, 0.147$ | $\mathbf{0.574 \pm 0.0933, 0.00523}$ |

Table 4: Weighted F-1 scores for various values of p across different classification methods.

| View | | Decision Tree | Naive Bayes | MLP | SVM | $k$-NN |
|---|---|---|---|---|---|---|
| Baseline | | $0.453 \pm 0.102, -$ | $0.372 \pm 0.099, -$ | $0.331 \pm 0.113, -$ | $0.48 \pm 0.211, -$ | $0.444 \pm 0.12, -$ |
| | p=0.25 | $\mathbf{0.55 \pm 0.0615, 0.00057}$ | $\mathbf{0.539 \pm 0.0831, 8.04e-07}$ | $0.389 \pm 0.129, 0.147$ | $0.383 \pm 0.0658, 0.0341$ | $0.532 \pm 0.111, 0.00562$ |
| | p=0.5 | $0.474 \pm 0.128, 0.439$ | $0.417 \pm 0.0687, 0.0306$ | $0.35 \pm 0.0103, 0.343$ | $0.381 \pm 0.0465, 0.0252$ | $0.437 \pm 0.118, 0.844$ |
| Enriched | p=1 | $0.433 \pm 0.101, 0.4$ | $0.354 \pm 0.0458, 0.377$ | $0.35 \pm 0.0103, 0.343$ | $0.35 \pm 0.0103, 0.00508$ | $0.45 \pm 0.0973, 0.798$ |
| | p=1.5 | $0.455 \pm 0.158, 0.925$ | $0.353 \pm 0.0234, 0.33$ | $0.384 \pm 0.101, 0.0318$ | $0.35 \pm 0.0103, 0.00508$ | $0.458 \pm 0.126, 0.611$ |
| | p=2 | $0.518 \pm 0.0882, 0.034$ | $0.367 \pm 0.0634, 0.734$ | $0.439 \pm 0.221, 0.0393$ | $0.569 \pm 0.114, 0.0123$ | $\mathbf{0.564 \pm 0.0894, 0.00129}$ |

## PARAMETERIZED BRAIN PLOTS

In this section we include additional plots from the feature weighting described in Section 5.4. These plots are for various values of $p$ and cover both the unsupervised and supervised model weighting to differentiate between all subjects and subjects with AD respectively. First, we include all the plots from the unsupervised feature extraction that denote the differences across all subjects. Second, we include the plots after weighting the biomarkers by the learned weights from the decision tree models with the best F-1 score at diagnosing patients with AD.

**Unsupervised plots**

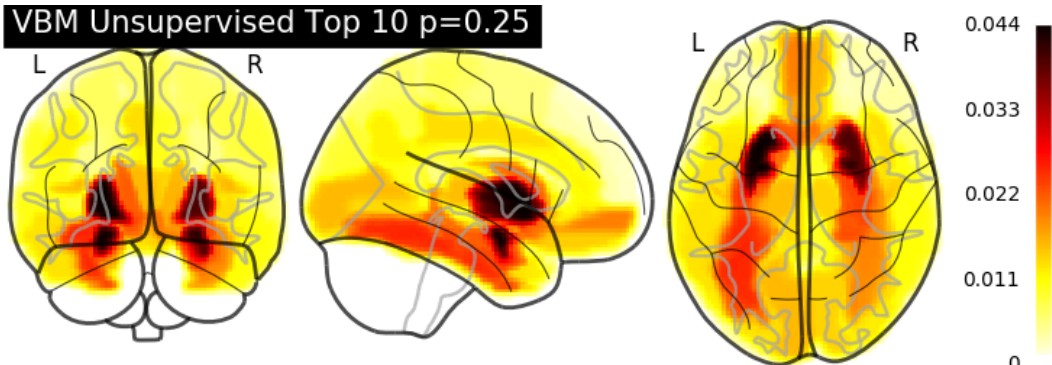

Figure 5: Unsupervised key biomarkers with $p = 0.25$.

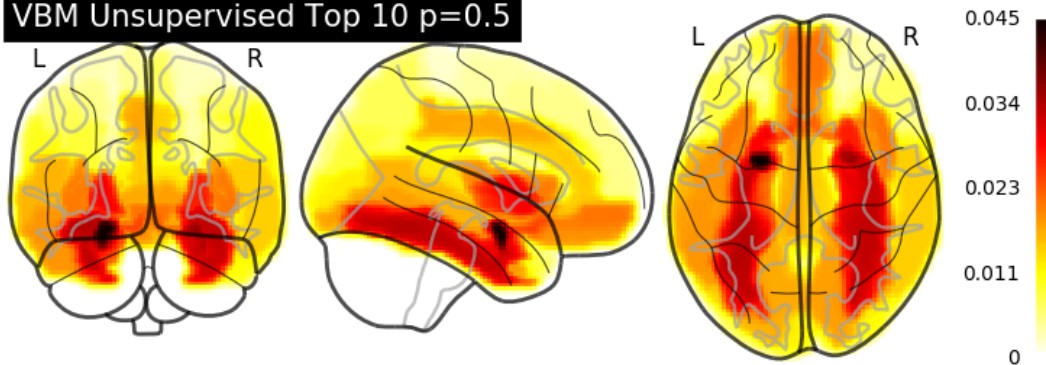

Figure 6: Unsupervised key biomarkers with $p = 0.5$.

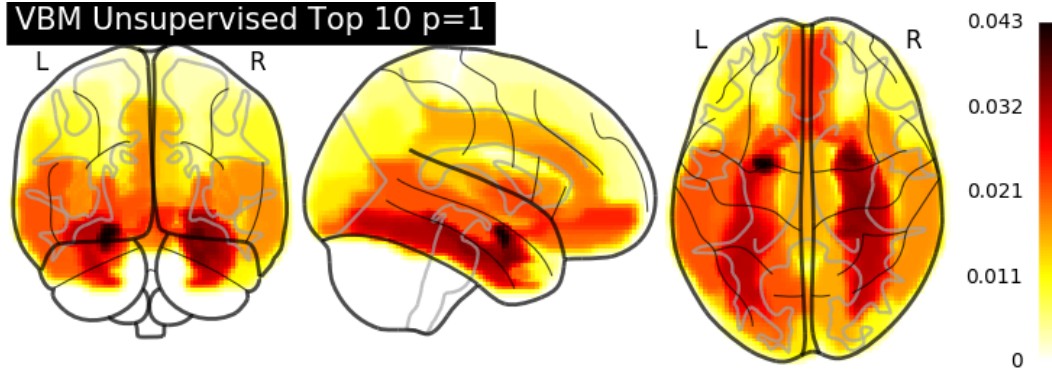

Figure 7: Unsupervised key biomarkers with $p = 1$.

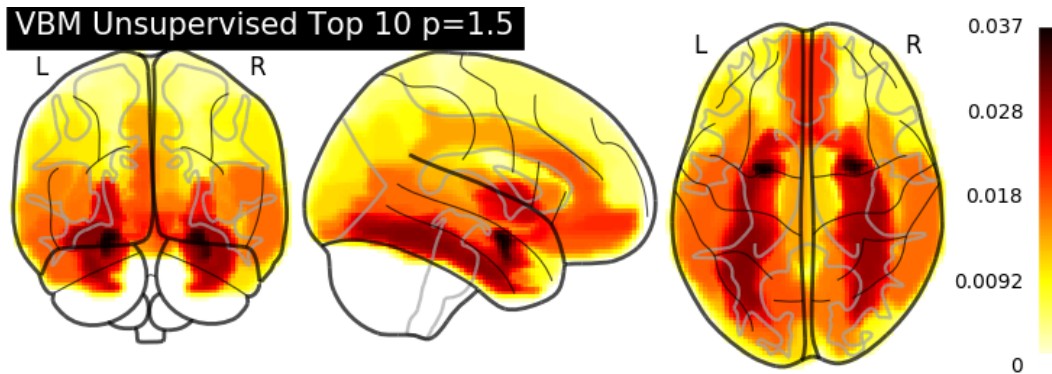

Figure 8: Unsupervised key biomarkers with $p = 1.5$.

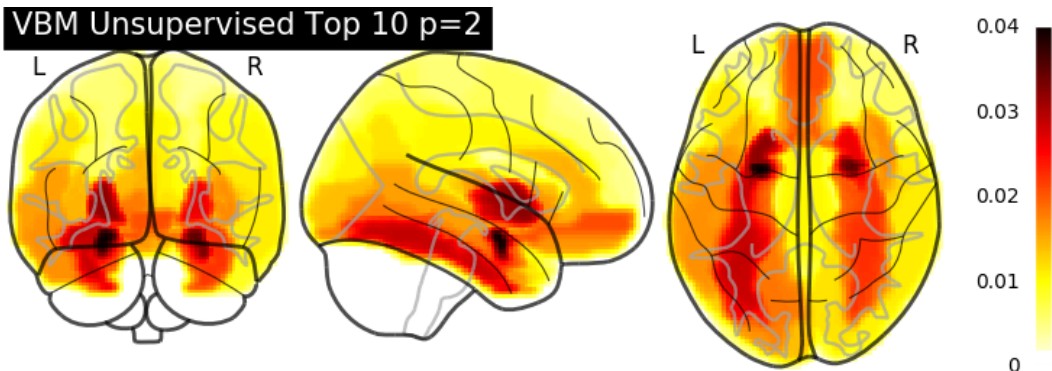

Figure 9: Unsupervised key biomarkers with $p = 2$.

**Supervised plots**

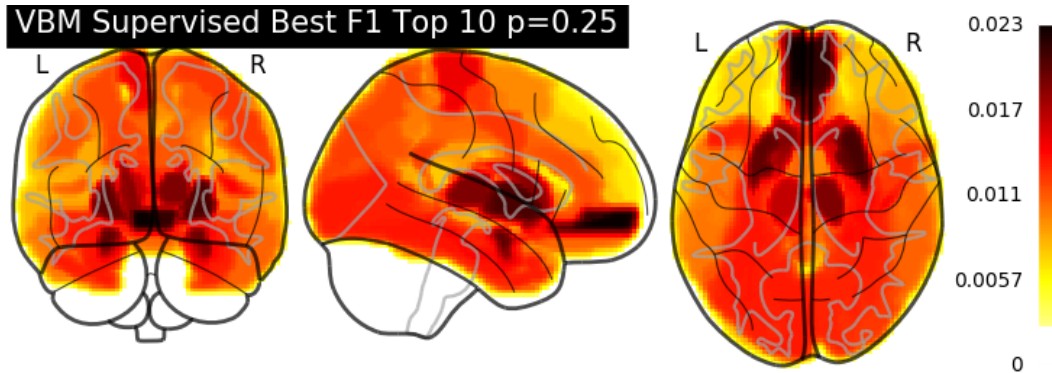

Figure 10: Supervised key biomarkers for AD diagnosis with $p = 0.25$.

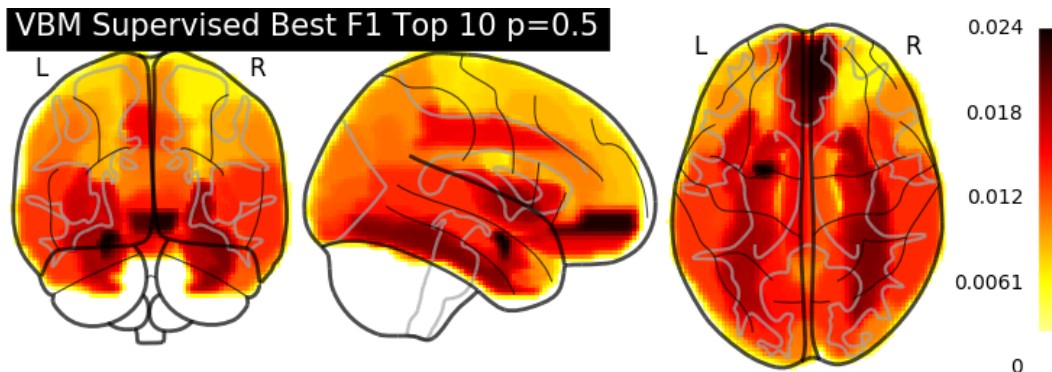

Figure 11: Supervised key biomarkers for AD diagnosis with $p = 0.5$.

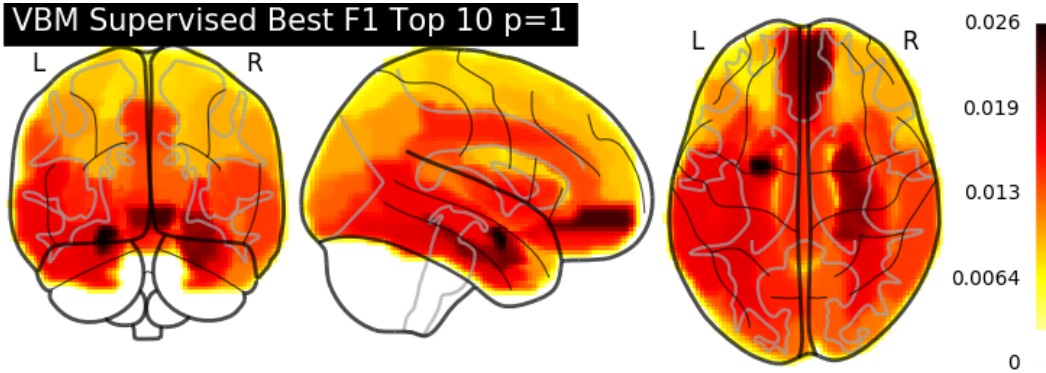

Figure 12: Supervised key biomarkers for AD diagnosis with $p = 1$.

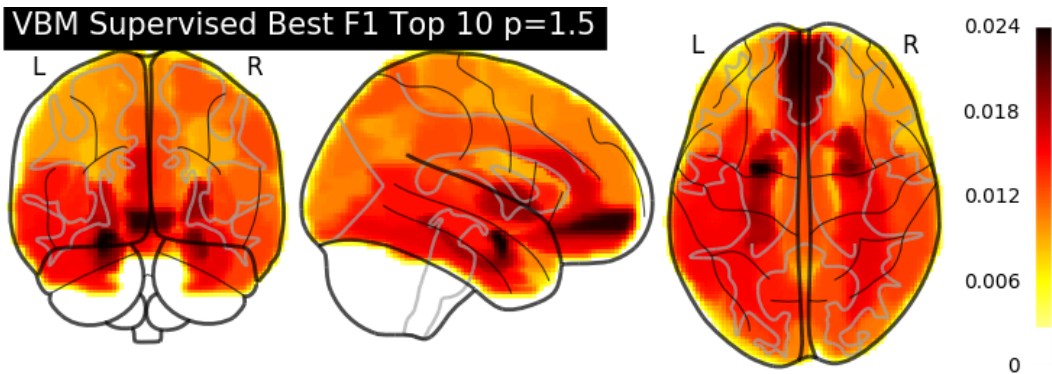

Figure 13: Supervised key biomarkers for AD diagnosis with $p = 1.5$.

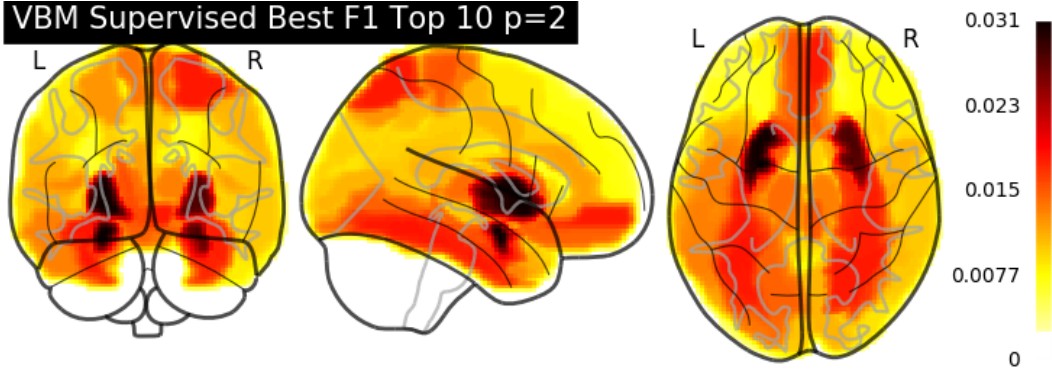

Figure 14: Supervised key biomarkers for AD diagnosis with $p = 2$.

