# OpenReview forum: "Longitudinal Enrichment of Imaging Biomarker Representations for Improved Alzheimer's Disease Diagnosis"
_ICLR.cc/2020/Conference — Reject_

### Official Review · AnonReviewer2 · 2019-10-24
**Official Blind Review #2**

**Rating:** 1

**Review:**

This manuscript develops a metric-learning with non-Euclidean error terms for robustness and applies in to data reduction to learn diagnostic models of Alzheimer's disease from brain images. The manuscript discusses a metric-learning formulation as a quotient for reconstruction-error terms, how to optimize the quotient based on results from Wang et 2014, an iterated reweighted approach to circumvent the non-smooth part of the l1 loss in zero, and experiments on brain images of Alzeimer's disease.

My two main issues with the manuscript is that the theoretical part is written very imprecisely and that the experiments are not convincing due to the lack of good baselines and of statistical power.

With regards to the theoretical contributions, a fraction of the results in the present manuscript are trivial consequences or Wang2014, and yet it comes with errors in the statements. For instance, in equation (9), the present manuscript writes greater or equal, while I believe that it should be strictly greater, as in Wang. Theorem 1 and 4 seem almost the same thing, though with a contradiction between the two. Other statements are inaccurate: the authors claim some results on reaching global optima, while I believe that they can only claim that they reach stationary points. Theorem 2 and 5 seem to be the same thing.

Concerning the iterated reweighted approach, I believe that this is non smooth only for g(x)=0, which is not covered by the theorems of Wang 2014. Is this algorithm needed? Note that Wang apply their algorithm with an l1 norm, ie non-smooth in zero, and do not report problems with out. The manuscript mentions that "to inverted matrices that divide 0s, which routinely lead to inferior learning performance.". I am not exactly sure what that means and I would need to understand better the problem. Also, the theoretical contribution that with the added the delta the algortihm converges, seems quite trivial: it seems to me that it is the eta trick.

Minor comments: in algorithm 3, it would be useful to write the full expression of the equations, rather than just reference the numbers. Also, the computational cost of the eigenvectors at each iteration seems quite prohibitive.

How was the value r=3 selected?

Figure 2 seem to choose that approaches have not converged: they final value is larger than intermediate values?

How were the p-values between cross-validation assessment of estimators computed? If it was done using standard paired t-test, this is incorrect are the folds are not independent.

With regards to the experiments, I worry that the model is not compared against simple baselines, such as a PCA.

**Experience Assessment:**

I have published in this field for several years.

**Review Assessment: Checking Correctness Of Derivations And Theory:**

I assessed the sensibility of the derivations and theory.

**Review Assessment: Checking Correctness Of Experiments:**

I assessed the sensibility of the experiments.

**Review Assessment: Thoroughness In Paper Reading:**

I read the paper thoroughly.

---

### Official Review · AnonReviewer1 · 2019-10-27
**Official Blind Review #3**

**Rating:** 3

**Review:**

The motivation of this work is to address the issue of inconsistent and missing data in longitudinal clinical studies.

The main goal of this work is to learn a dimensionality reduction representation by combining locality preserving projection (LPP) and and the global pattern by principal component analysis (PCA). The balance between these two components is modulated by a hyperparameter alfa. The authors proposed a method to overcome the issue of a direct computation of alfa. The main contribution of this work is to prove that a quotient formulation between LPP and PCA is equivalent to a weighted difference of these two terms.

It is not straightforward to understand whether the proposed method is effective in reducing the time consuming effort required by the computation of alfa hyperparameter, as stated in the premises as motivation of this work.

The presentation of theorems is misleading. Theorems 1 and 4 seem to claim opposite statements, while theorems 2 and 5 have the same claim.

Minor comments.
After equation 5 there is a typos in the ratio LPP/LPP.


**Experience Assessment:**

I have read many papers in this area.

**Review Assessment: Checking Correctness Of Derivations And Theory:**

I did not assess the derivations or theory.

**Review Assessment: Checking Correctness Of Experiments:**

I assessed the sensibility of the experiments.

**Review Assessment: Thoroughness In Paper Reading:**

I read the paper at least twice and used my best judgement in assessing the paper.

---

### Official Review · AnonReviewer3 · 2019-11-04
**Official Blind Review #3**

**Rating:** 1

**Review:**

This paper proposed an unsupervised method to make better usage of the inconsistent longitudinal data by minimizing the ratio of Principal Components Analysis and Locality Preserving Projections.

Comments:
1. It is out of the area of my expertise, but the innovation of combining the PCA and LPP to produce the dimensionally reduced representation of the longitudinal data seems not significant enough, especially when the author did not mention why they chose these two methods out of all the available projection/hashing approaches.

2. The paper is poorly written. Theorem 1 and Theorem 4 state the opposite conditions for v. Theorem  2 and Theorem 5 are exactly the same. If the same theorem has already been proved by previous work [Wang et al., 2014], it is not necessary to prove it again in this paper. The most important theorem which is claimed as the theoretical contribution of this paper (i.e. Theorem 6) did not even appear in the main contents but only in the appendix. The proof of theorems in appendix, Proof A.1, A.2, A.3 did not state which theorem they are corresponding to and are organized in different order as Theorem 3-5, make it hard to read.

3. There are many grammar errors and sentences which are not consistent with English writing conventions. E.g. “This problem is heavily studied with (Wang et al., 2012; Brand et al., 2018; Lu et al., 2018) as example approaches which, despite their effectiveness, present an added complexity to the problem.” is better written as “This problem is excessively studied by (Wang et al., 2012; Brand et al., 2018; Lu et al., 2018) as example approaches which, despite their effectiveness, present an additional complexity to the problem”; “... to provide value in predicting patients’ diagnosis.” is better written as “... to provide valuable information in ...”; “... , we focus AD diagnosis from MRI scans ...” should be “... , we focus on AD diagnosis from MRI scans ...”; “consider outliers” is better as “handle outliers”; etc.

4. In the sentence below equation (5), “J_LPP/J_LPP” should be “J_LPP/J_PCA”.


**Experience Assessment:**

I do not know much about this area.

**Review Assessment: Checking Correctness Of Derivations And Theory:**

I assessed the sensibility of the derivations and theory.

**Review Assessment: Checking Correctness Of Experiments:**

I assessed the sensibility of the experiments.

**Review Assessment: Thoroughness In Paper Reading:**

I read the paper at least twice and used my best judgement in assessing the paper.

---

### Decision · Program_Chairs · 2019-12-19

**Decision:**

Reject

**Comment:**

This paper proposes to overcome the issue of inconsistent availability of longitundinal data via the combination of leveraging principal components analysis and locality preserving projections. All three reviewers express significant reservations regarding the technical writing in the paper. As it stands, this paper is not ready for publication.